# Climate extremes in Svalbard over the last two millennia are linked to atmospheric blocking

Francois Lapointe [1,2] ✉, Ambarish V. Karmalkar [3], Raymond S. Bradley [1,2], Michael J. Retelle [4,5] & Feng Wang [6]

Arctic precipitation in the form of rain is forecast to become more prevalent in a warmer world but with seasonal and interannual changes modulated by natural modes of variability. Experiencing rapid hydroclimatic changes in the Arctic, Svalbard serves as an ideal study location due to its exposure to oceanic and atmospheric variability in the North Atlantic region. Here we use climate data from paleoproxies, observations, and a climate model to demonstrate that wet and warm extremes in Svalbard over the last two millennia are linked to the presence of atmospheric blocking regimes over Scandinavia and the Ural mountain region. Rainfall episodes lead to the deposition of coarse sediment particles and high levels of calcium in Linnévatnet, a lake in south-west Svalbard, with the coarsest sediments consistently deposited during atmospheric blocking events. A unique annually resolved sediment record from Linnévatnet confirms that this linkage has been persistent over the past 2000 years. Our record also shows that a millennial-scale decline in Svalbard precipitation ended around the middle of the 19th century, followed by several unprecedented extreme events in recent years. As warming continues and sea ice recedes, future Svalbard floods will become more intense during episodes of Scandinavian and Ural blocking.

Temperatures across the Arctic are rising at a rate of 2–3 times the global average[1] with some parts of the region experiencing even more rapid changes. In particular, the Svalbard archipelago, halfway between Norway and the North Pole, is at the leading edge of climatic changes taking place in the Arctic. Over the past century, temperatures in Svalbard have undergone a concerning increase of approximately 4 °C[2]. Particularly alarming is the fact that since 1991, this region has experienced a warming trend that surpasses the already elevated Arctic temperature average more than twofold[2]. This rapid increase is linked to the strong flux of warm Atlantic water and associated warm air that nowadays commonly extends northward into the western fjords of Svalbard and even penetrates well into the Arctic Ocean[3,4].

Furthermore, Svalbard is located in the path of a strong poleward flow of moisture in the form of atmospheric rivers in the North Atlantic sector, which drives sea ice decline and precipitation extremes in the region[3,4]. Considering that rapid air and ocean temperatures have increased, and sea ice has declined around the Nordic Seas in recent decades, Svalbard serves as an ideal natural laboratory to study drivers of long-term hydroclimate variability, shedding light on the impending changes that can be expected across in the wider Arctic region.

Recent observations suggest that rapid hydroclimatic changes may be already underway in Svalbard. During the 2015–16 winter, the seas around Svalbard were exceptionally warm and almost completely ice-free on several occasions, leading to above-normal precipitation in

[1]Department of Earth, Geographic and Climate Sciences, University of Massachusetts, Amherst, MA, USA. [2]World Climate Research Programme - Climate and Cryosphere (CliC) Project, University of Massachusetts, Amherst, MA, USA. [3]Department of Geosciences, University of Rhode Island, Kingston, RI, USA. [4]Earth and Climate Sciences, Bates College, Lewiston, ME, USA. [5]Department of Geology, The University Center in Svalbard, Svalbard, Norway. [6]Institut National de la recherche scientifique, University of Québec, Québec, QC, Canada. ✉e-mail: flapointe@umass.edu

Svalbard. At Kapp Linné, total precipitation from June-August, 2016 was 113 mm, compared to the long-term average of 72 mm based on measurements since 1912[5]. Then in October 2016, when temperatures are normally well below freezing, more than 42 mm of rain fell in a single day[6]. As the active layer was still largely unfrozen, the ground quickly became saturated, which had dramatic consequences for the stability of slopes in the region and resulted in hazardous mudflows, leading to the evacuation of parts of the town of Longyearbyen. In February 2017, heavy rainfall and above freezing temperatures resulted in slushflows and rivers running over frozen ground. The rain-on-snow event was followed by an avalanche that destroyed several homes in the town. These types of unprecedented events represent a paradigm shift in the hydroclimatic regime, posing new dangers to people and infrastructure, and to wildlife in the region[7–9].

While thermodynamics can help explain the long-term Arctic-wide projections, understanding of the dynamics, especially the short-term climate variability and associated circulation changes, is critical for evaluating hydrological extremes at regional scales. In this respect, atmospheric blocking plays a crucial role in regulating moisture transport within the northernmost reaches of the North Atlantic current, including the Arctic[10]. There are three circulation patterns that tend to occur more frequently for low Arctic sea ice conditions, namely the Scandinavian, Ural and Greenland Blocking patterns[11–16]. During episodes of Greenland Blocking, characterized by a negative North Atlantic Oscillation (NAO) pattern, warm and moist air is advected from lower latitudes towards Greenland[17]. The intensification of Greenland Blocking since 2005 has played a significant role in the recent acceleration of surface melting over the Greenland Ice Sheet, as well as a reduction in the area of Arctic sea ice[18]. Scandinavian and Ural blocking patterns also result in the advection of warm air masses from mid-latitudes with positive temperature anomalies of up to 8 °C in the region of Svalbard[11]. Blocking episodes thus have a strong influence on regional climatic conditions in the Arctic and yet changes in their character, persistence, and evolution as the overall climate warms remains an open question. Several factors impede a comprehensive understanding of blocking episodes and their impacts: the insufficient length of instrumental data, the absence of proxies capable of describing blocking events before the advent of instrumental records, and the variable depiction of blocking in the current generation of climate models[19–22].

In this paper, we provide a long-term perspective on hydroclimate changes and their drivers in Svalbard and the neighboring Arctic region by analyzing observations (instrumental data), paleoproxies (lacustrine sediments and tree-rings) and a paleoclimate model simulation. We first examine the synoptic conditions responsible for recent extreme events in Svalbard using instrumental data, then examine how these relationships have varied over the past two millennia using a paleoclimate simulation and an annually laminated sedimentary record from Lake Linné (Linnévatnet) in western Svalbard. Specifically, we examine the role of atmospheric blocking over Scandinavia in driving temperature and precipitation variability in Svalbard and discuss the potential role of sea ice and sea surface temperature changes in the Nordic Seas.

## Results

### Atmospheric drivers of extreme precipitation

The extreme precipitation events in Svalbard in 2016 resulted in one of the wettest June through November (Jun-Nov) periods since records began in 1955 (Fig. 1a). In general, wet conditions in Svalbard are associated with positive mid-tropospheric geopotential height anomalies south and southeast of the archipelago and warmer than normal conditions across the region. The regression between monthly precipitation in Svalbard between Jun–Nov and 500 hPa geopotential height (Z500) anomalies over the modern period (see Methods) shows a high-pressure system centered over northern Scandinavia (Fig. 1b), a

circulation feature which has been described as the "Scandinavian blocking pattern" (SB)[23–25]. The SB causes diversion of the typical west-to-east flow of the polar jet stream, and the development of a tilted jet structure from southwest to northeast[26,27], with lower pressure over Greenland. That creates ideal conditions for the advection of moisture towards Svalbard[27], as evident in the regression maps between Svalbard Jun–Nov precipitation and geopotential height and wind anomalies at 500 hPa (Fig. 1b). These wet months in Svalbard are also associated with higher SST anomalies in the Nordic Seas (Fig. 1c) and high surface air temperatures in the region which includes all of Scandinavia and the Arctic Ocean north of Siberia (Fig. 1d). This relationship between high rain events in Svalbard and atmospheric blocking over Scandinavia prevails regardless of the season, even during winter (Table S1, Figs. S1, S2). Overall, atmospheric blocking patterns can persist for several days or even weeks, leading to prolonged periods of relatively dry and sunny weather in parts of Northern and Central Europe, colder temperatures in the eastern parts of the Eurasian continent, and wet conditions in Svalbard[28].

### Lake sediment analysis

Limnological and sedimentological measurements have been made in Linnévatnet (Fig. 2) since 2004, and the sediment traps reveal that sedimentation occurs when the lake is ice-free, essentially from Jun-Nov. Intervalometers deployed in the lake have recorded sedimentation at half hourly intervals, providing a unique temporal record of sediment fluxes into the lake. The coarsest grain-size deposition as well as high calcium (Ca) values consistently occurred during periods of heavy precipitation and SB due to increased stream flow on the alluvial fans that drain from the eastern carbonate bedrock terrain. Figure 2e shows an example of sediment grain-size from 07/2015–07/2016. The coarsest grain-size and highest sediment fluxes occurred on September 11, 2015 and May 28, 2016 when the Z500 anomaly for these two days clearly shows the presence of strong SB (Fig. 2e–g). Similarly, on October 15 2016, -81 mm of sediment was deposited in the intervalometer receiving tube (overfilling the sediment trap) and there was a strong SB pattern on that day as well (Fig. S3). Other data available from the intervalometers indicate that days recording the coarsest sediments contribute to most of the annual sediment fluxes and that all those days are characterized by a blocking anomaly either over northern Scandinavia or Eurasia along with warmer SSTs in the Nordic Seas (Figs. S4–S16). Therefore, whether examining monthly (Fig. 1) or daily anomalies (Figs. 2, S1–S16), the underlying atmospheric dynamics remain consistent, primarily driven by atmospheric blocking over Scandinavia and the Ural region.

### Svalbard hydroclimate over the last 2 millennia

Linnévatnet contains a unique annually laminated (varved) sediment record[29] (Supplementary Note 1; Fig. S17) and provides the opportunity to investigate the long-term association between extreme precipitation events and Scandinavian Blocking. The primary source of water for Linnéelva (the main river system draining into the lake) originates predominantly from the melting snow and the Linnébreen glacier, a small valley glacier situated 7 kilometers to the south of Linnévatnet[30]. The melting of the local glacier Linnébreen contributes to the accumulation of coarser sediments, and its size influences the grain-size variability in the lake sediments. In addition to being a temperature proxy, the grain-size (D50 μm) is also influenced by large rain events whereby increased rainfall induce coarser sediments; it is thus a mixture of temperature and precipitation variability[29]. Samples collected around the lake's watershed have shown that positive micro-fluorescence (μ-XRF) Ca anomalies were mainly found in the carbonate platform from the eastern valley[29]. Hence, high levels of calcium, often accompanied by a higher proportion of coarse particles (Supplementary Note 2, Fig. S18)[29,30], results from increased stream flow through the eastern carbonate bedrock terrain resulting from rainfall

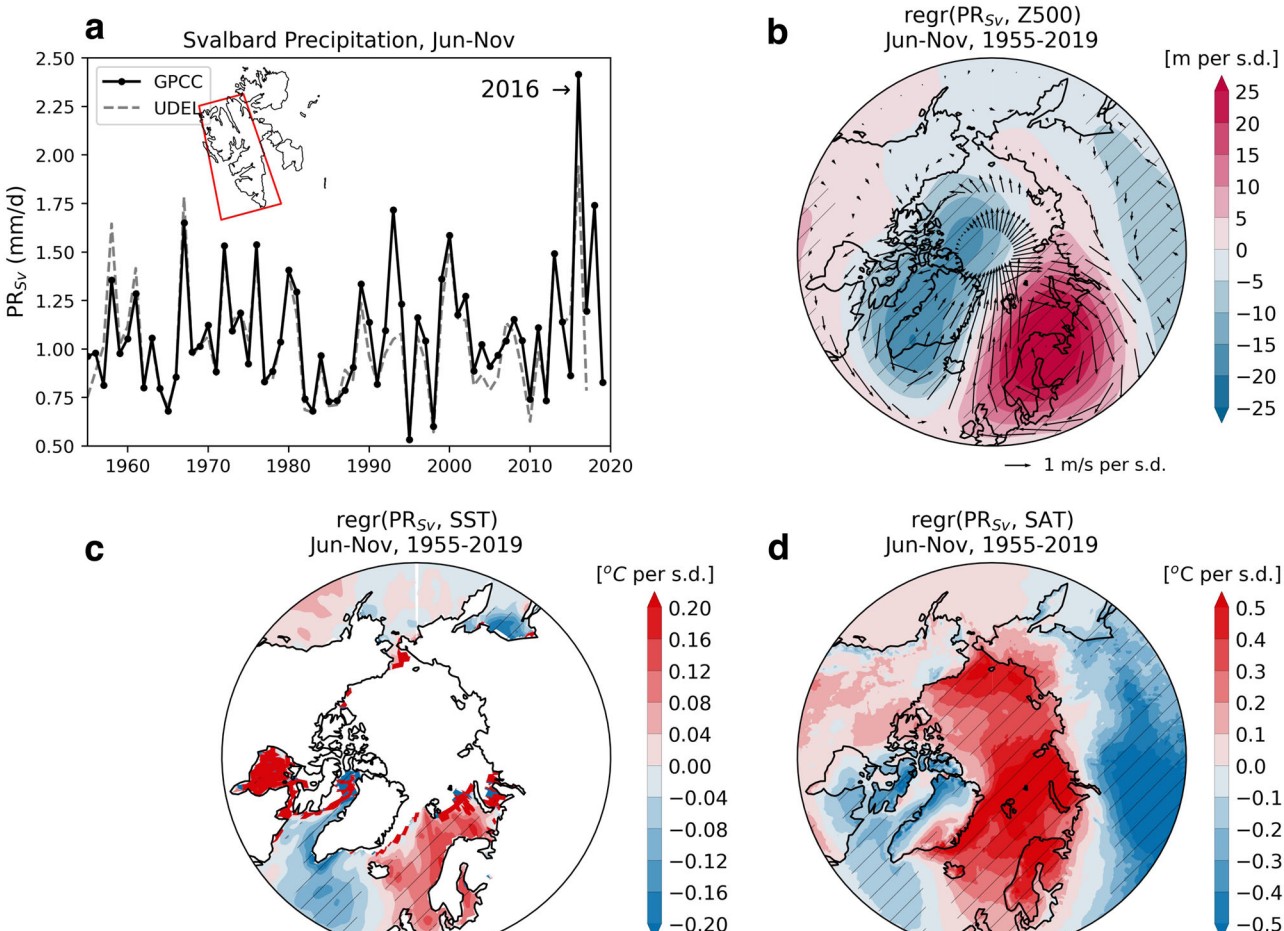

**Fig. 1 | The relationship between Jun–Nov precipitation in Svalbard and atmospheric circulation features. a** Seasonal mean precipitation in Svalbard ($PR_{Sv}$; red rectangle in the inset) from June through November in two land-only observational datasets (see Methods): GPCC (solid black) and UDEL (dotted grey). Regression maps showing the relationship, for the period 1955–2019, between Jun–Nov monthly precipitation (GPCC $PR_{Sv}$) and (**b**) 500 hPa geopotential height (Z500) anomalies (colors) and winds (V500; arrows) from ERA5; (**c**) Sea surface temperature (SST) anomalies from HadISST dataset and (**d**) surface air temperature (SAT) anomalies from ERA5. Regression maps show changes for one standard deviation change in the monthly $PR_{Sv}$ time series. Cross-hatching shows regions with significant regression coefficients at the 95% significance level.

events. Ca input to the lake is a particularly good proxy for medium to large rainfall events (>10 mm day$^{-1}$), which is less evident in the grain-size measurements (Fig. S18). Calcium variability is therefore considered as a reliable proxy for tracking rainfall events, with less influence resulting from changes in the glacier's size compared to grain-size alone.

Fluctuations in Ca show a long-term decreasing trend from the Medieval period to the Little Ice Age (LIA), with lowest values centered from the 1600s to the mid-1800s (Fig. 3a). This declining trend reversed in the ~1850s when values started to rise steadily. The record also indicates that the floods in 2016 were certainly exceptional in the context of the last ~1600 years[29]. Figure 3a compares μ-XRF Ca data from Linnévatnet sediment cores with a 1250-year summer temperature reconstruction over Scandinavia, based on a spatial climate reconstruction from absolutely dated tree-ring records[31]. Zhang et al.[32] found that summer SB was associated with extremely warm temperatures in the Fennoscandia/Scandinavia region as expressed in tree rings by increased latewood density. Clear skies during persistent summer anticyclones result in higher temperatures, which benefit tree growth[32]. There is significant co-variability between the Scandinavian reconstruction and Linnévatnet Ca over their overlapped period (Fig. 3a, $r = 0.30$, annual, 5-year running mean on tree rings $r = 0.42$,

$p < 0.001$). Our Ca record is further significantly correlated to other tree-ring based summer temperature reconstructions from Scandinavia (Torneträsk in Northern Sweden) and the Eurasian regions spanning the last 1500–2000 years[33,34] (Fig. S19).

The spatial correlation between precipitation in Svalbard and Z500 (Fig. 1b) exhibits a similar pattern to that between Z500 anomalies and the composite tree-ring temperature reconstruction for Scandinavia (cf. Fig. 3b). Tree rings from Scandinavia are also strongly and positively correlated with SST anomalies in the North Atlantic, with greatest correlation in the northern seas extending to Svalbard (Fig. 3d), a pattern which is strikingly similar to that between SSTs and Svalbard precipitation (cf. Fig. 1c). Overall, the region extending from 20°W to about 80°E – 60°N to 80°N is much warmer during strong SB.

There are, however, some differences between these datasets which may be due to the fact that high-latitude tree rings are essentially summer proxies whereas our sediment record is more sensitive to both summer and fall precipitation[29]. Other differences may arise due to the presence of sea ice southwest of Svalbard, creating a cooler boundary layer that would lift the moisture-laden warm air upward resulting in more precipitation over the Greenland Sea[35]. This process can reduce the amount of moisture reaching Svalbard even when there

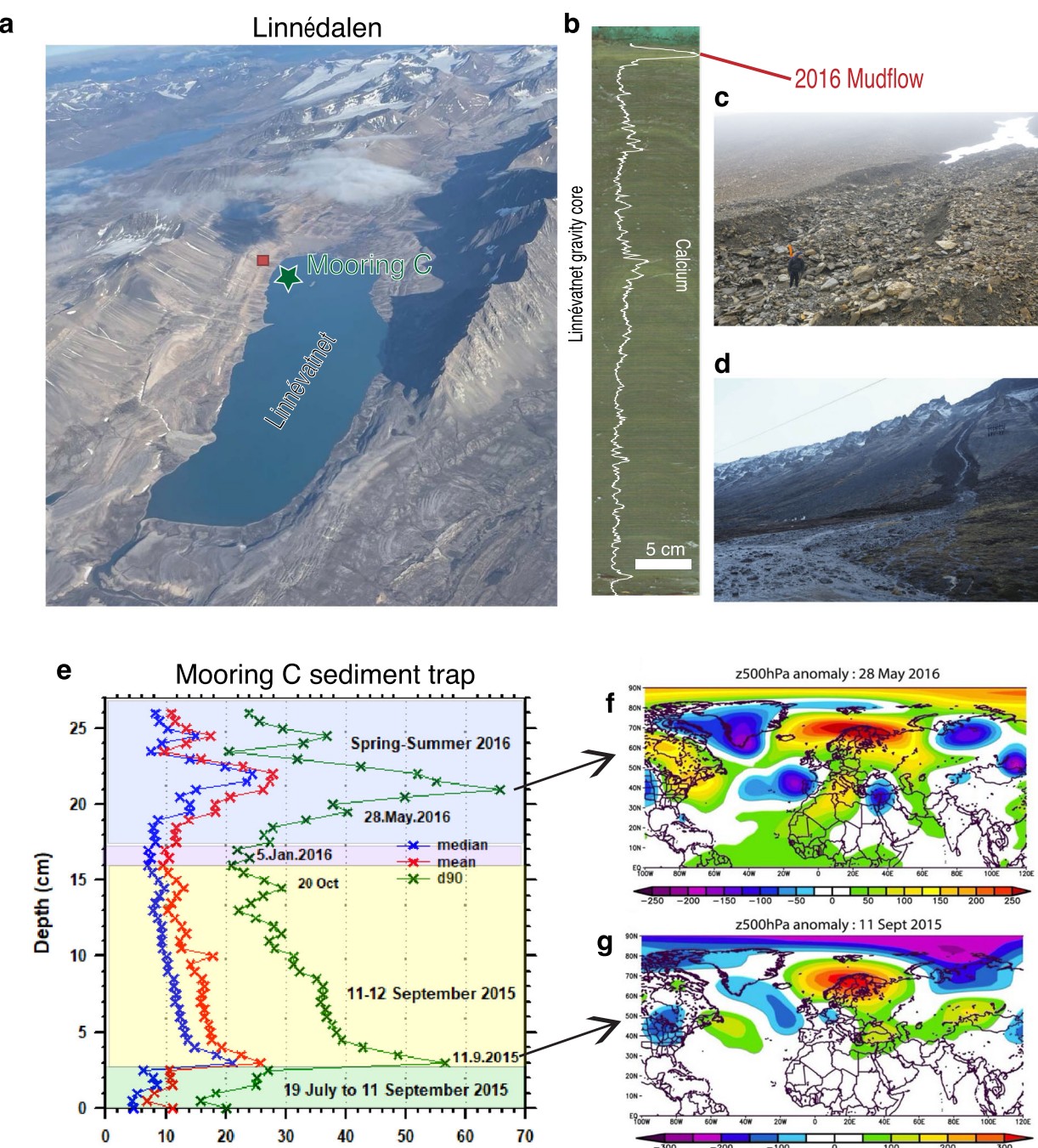

**Fig. 2 | Linnévatnet's sediment traps reveal the influence of atmospheric blocking in regulating flooding events. a** Air photo of Linnévatnet and location of intervalometer at mooring C (coring site). **b** Photograph of the gravity sediment core collected in 2019 showing the μ-XRF calcium variation throughout the first 41 cm. The year 2016 is characterized by highest Ca values. **c** Scarp on the eastern carbonate bedrock terrain (red square in (**a**)) triggered by anomalously warm weather and intense rainfall recorded on October 15, 2016. The worst mudflow in 40 years occurred that year with roads being completely flooded in Longyearbyen ((**d**), photograph from W.R. Farnsworth)[6]. **e** Grain-size data collected from the sediment trap at mooring C in 2015–2016 showing two days with coarsest grain-size peaking on September 11, 2015 and May 28, 2016, respectively. The background colors are from bottom to top: Green = summer 2015, yellow = fall 2015, purple = winter 2015/2016 and blue = spring and summer 2016. The sample symbols Blue = median, d50, red = mean and green = d90. **f–g**, Atmospheric pressure anomalies at 500 hPa Geopotential height (Z500) (relative to 1980–2010) during those two days calculated from daily National Centers for Environmental Prediction reanalysis data.

is a strong SB pattern. The comparison between Calcium variability and the reconstructed sea ice concentration (SIC) from the Nordic Seas[36] reveals a significant correlation (r = −0.37, annual, 5-year running mean on sea ice r = −0.46, p < 0.001) over the past ~800 years (Fig. 3c). Lower SIC persisted from the 1200s to the late 1500s,

followed by a sudden and substantial increase in sea ice that endured until the mid-1800s (Fig. 3c). Consequently, the sustained low Ca values during this cold period likely stem from a persistent positive sea ice conditions in the area, leading to a decline in rainfall events in Svalbard.

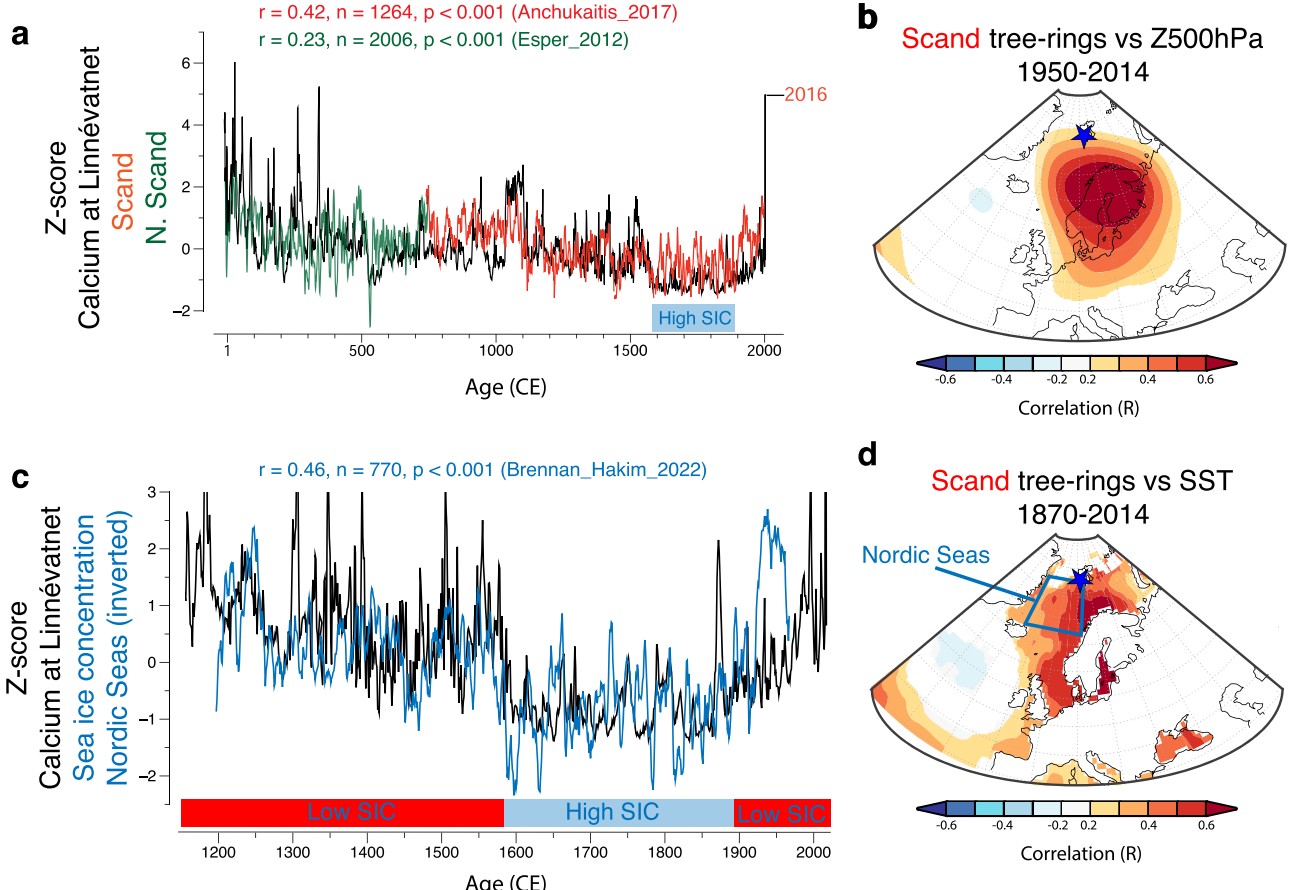

**Fig. 3 | Paleo evidence links Scandinavian blocking to Common Era flood events in Svalbard. a** The Scandinavian (Scand) reconstructed summer temperatures from the N-TREND tree-ring dataset (in red; data were area-weight averaged over 12–30°E and 65–70°N)[31] and the Northern Scandinavian reconstructed temperature (in green; N. Scand) from Esper et al.[60] compared to the Calcium variability at Linnévatnet (in black). For visibility only the 1-750CE time interval is shown for the N. Scand[60] ((**a**); green) as the two Scand reconstructions exhibit strong co-variablity during their overlapping period (750 CE-2006 CE, annual: r = 0.87). **b** Spatial correlation between the Scand temperature reconstruction[31] and atmospheric pressure at 500 hPa geopotential height (Z500)[61]. The blue star denotes the location of Linnévatnet in Svalbard. **c** Reconstructed sea ice concentration from the Nordic Seas[36] (blue) compared to the Calcium at Linnévatnet. Periods of lower and higher sea ice concentration (SIC) are highlighted in red and blue, respectively. **d** Spatial correlation between the Scand temperature reconstruction[31] and sea surface temperature from HadISST. Blue area delimits the location of the targeted reconstructed SIC from the Nordic Seas in (**c**). The tree-ring and sea-ice time series were filtered using 5-year running mean to improve visibility. Correlations shown (**b**, **d**) are significant with a P < 0.1.

## Simulations of past and future hydroclimate

To provide a long-term perspective on the presence and persistence of the SB circulation pattern, we evaluated the MPI-ESM1-2-LR *past2k* transient simulation (1–1850 CE)[37], carried out as part of the Paleoclimate Model Intercomparison Project Phase 4 (PMIP4). First, we calculate the SB index following methods discussed in Lee et al.[38]. The principal component analysis of the regional Z500 anomaly field (see Methods) shows that NAO is the leading mode of variability followed by the mode representing the SB pattern (described as Scandinavian-Greenland pattern in Lee et al.,[38]). This decomposition of Z500 variability is similar to that over the modern period[38], clearly indicating that the Scandinavian blocking regime has been a persistent circulation feature in the region over the last two millennia. Regression maps between the SB index, described by the second principal component time series, and Arctic-wide Z500 and SAT anomalies indicate the presence of a broad-scale pattern (Fig. 4a) and associated warm conditions in Svalbard and over Scandinavia (Fig. 4b). Regressing the Z500 anomaly field on simulated Svalbard precipitation time series also produces a pattern that is strikingly similar to the SB pattern (Fig. 1), clearly illustrating the close link between blocking and precipitation in the Svalbard region.

Model simulations indicate a positive relationship between temperature and precipitation where warmer months between Jun-Nov are wetter in Svalbard (r = 0.49; Fig. 4d), and vice versa. As expected, we also find strong correspondence between the SB index and the southerly low-level moisture flux (qv850) over the Greenland Sea (r = 0.52; Fig. 4c, e). Focusing on the extremes, the top 1% of the wettest and warmest Jun-Nov months in Svalbard are associated with high values for both, the SB index and qv850 over the last two millennia. Thus, there is model evidence that temperature and precipitation anomalies in Svalbard are strongly linked to the variability in the SB pattern and the associated moisture transport to the region. Note, however, that there are instances in which strong SB pattern does not result in high precipitation in Svalbard and that some wet episodes occur when the SB pattern is weak or absent (Fig. 4d, e). This points to the role of other factors, such as sea ice, SSTs, and other moisture sources. As mentioned above, the presence of sea ice in the Nordic Seas would limit moisture transport towards Svalbard. A good example is the prolonged high SIC during the late Little Ice Age (~1600–1850), documented in the Nordic Seas (Fig. 3c), which likely prevented rain occurrences in Svalbard, despite the positive SB conditions. Nevertheless, these findings underscore that there has been a persistent association between the climate variability in Svalbard and atmospheric blocking over Scandinavia throughout the last two millennia.

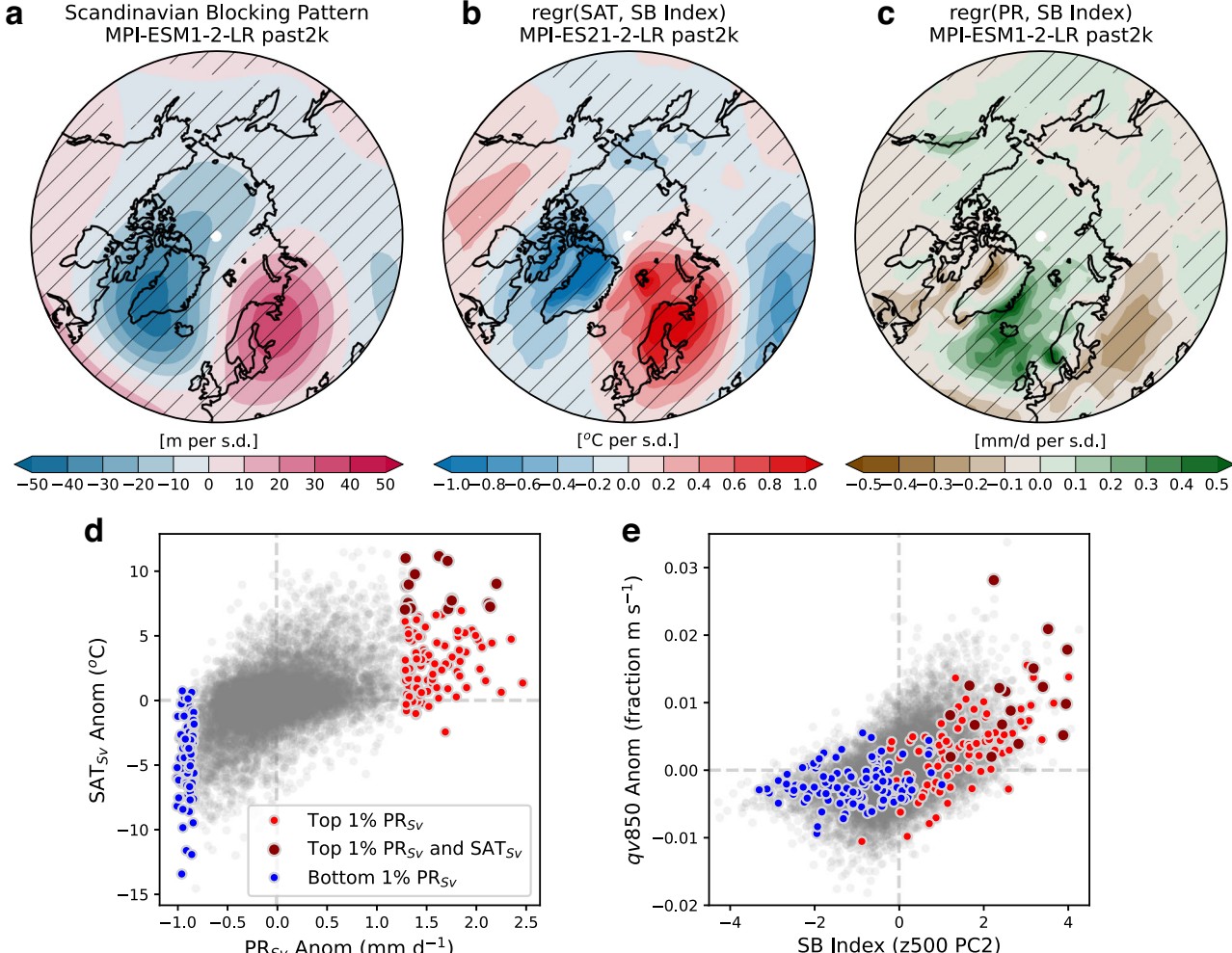

**Fig. 4 | Relationship between regional circulation and Svalbard temperature and precipitation in MPI-ESM1-2-LR transient simulation over the Common Era (*past2k* experiment). a** Scandinavian Blocking (SB) pattern calculated by regressing Jun-Nov monthly 500 hPa geopotential height (Z500) anomaly field on the SB index (see Methods). **b** Regressions of Jun-Nov monthly temperature and the SB index. **c** Regressions of Jun-Nov monthly precipitation and the SB index. **d** Relationship between Jun-Nov monthly temperature and precipitation in over the period 1–1850 CE. The spatial average is calculated over the domain shown in Fig. 1a. **e** Relationship between the strength of the SB index and low-level moisture transport over the Greenland Sea. Cross-hatching shows regions with significant regression coefficients at the 95% significance level.

We then use two climate models from the Coupled Model Intercomparison Project Phase 6 (CMIP6) archive, that span a wide range of temperature and precipitation projections for the Arctic, to demonstrate that the recent trend toward wet conditions in Svalbard is projected to persist in the future under scenarios of continued warming. Figure 5a shows an increase in mean Jun-Nov tempeature and precipitation over the period 2015–2099 in Svalbard across three emissions scenarios. While the magnitude of precipitation increase is dependent on the model structure and future emission trajectories, these results clearly suggest a wetter climate in Svalbard with future warming. Notably, this increase in seasonal precipitation is accompanied by an increased frequency of extreme rainfall in the region, as indicated by the region experiencing more days with rainfall >10 mm by the end of this century (Fig. 5b). However, connecting this projected increase in precipitation with changes in the character of blocking episodes under future warming scenarios remains challenging. The representation and projections of atmospheric blocking in the current generation of models are areas of active research with considerable uncertainties[19,20,39]. Therefore, further analysis using daily data from multiple climate models and emissions scenarios will be necessary to assess the robustness of future extreme precipitation projections and their relationship with atmospheric blocking.

## Discussion

In the last four decades, there has been a notable uptrend in atmospheric blocking, resulting in a enhanced poleward energy transport, with significant decline of sea ice concentration as a response[10]. We find evidence in proxy data and model simulations that climate excursions in the Greenland-Eurasian sector of the Arctic are mainly driven by atmospheric variability rather than regional SST variations. This is exemplified by the analysis of occurrences of heavy rainfall which revealed the SB weather pattern as the predominant factor behind impactful precipitation events in Svalbard. We observe instances where blocking extends from the Scandinavian region through western Siberia, with the Urals at its focal point (Ural Blocking), exhibiting greater persistence during the fall versus summer season (Fig. S20). While certain events may not fit the mold of a typical SB (Figs. S1a, b, i, S2c, e, i), they nevertheless exhibit anticyclonic circulation over northern Scandinavia and Eurasia. Hence, strong relationship between Svalbard precipitation and regional SSTs found in proxy data may simply arise from the dominant role of blocking patterns on both regional temperature, sea ice variations and precipitation.

The sediments in Linnévatnet provide a unique temporal perspective on when blocking over Scandinavia was active over the past ~2000 years. In this respect, the abrupt decline in Ca values towards the

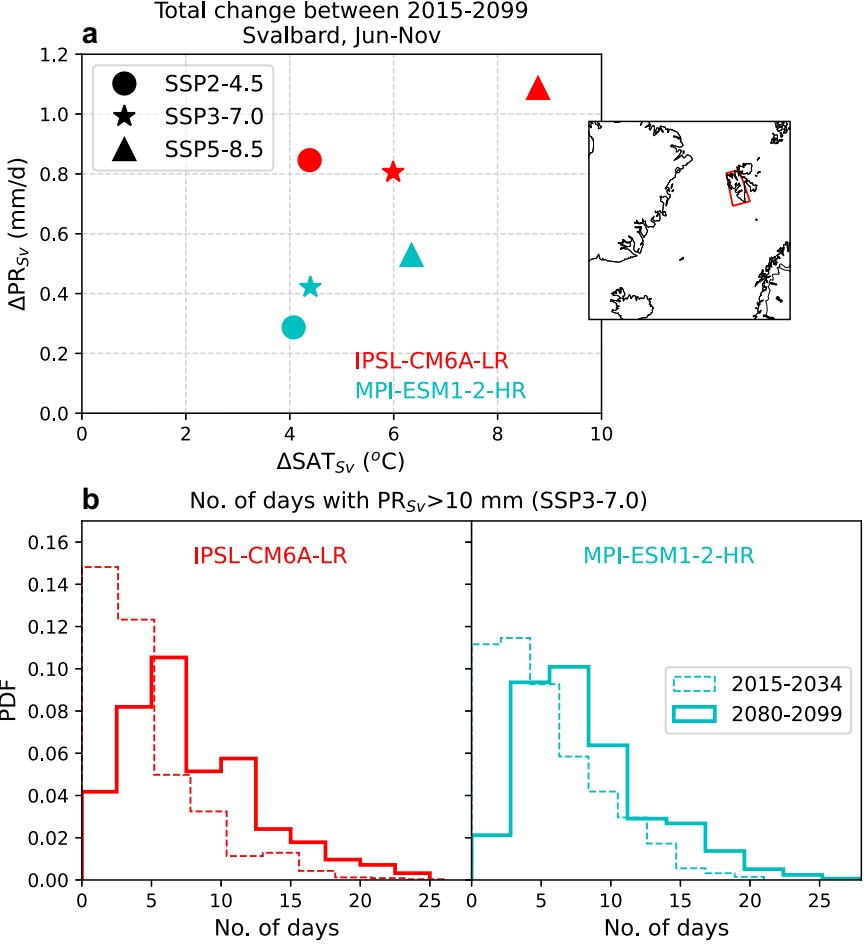

**Fig. 5 | Climate projections for Svalbard based on two CMIP6 models. a** Total change in Jun-Nov mean temperature and precipitation in Svalbard (see inset for area definition) between 2015 and 2099 based on linear trends for three emissions scenarios (SSP2-4.5, SSP3-7.0, SSP5-8.5). **b** Histograms showing the distributions of number of days with precipitation >10 mm across Svalbard (at 0.5° spatial resolution) over the period 2015–2034 (dashed line) and 2080–2099 (solid line) under SSP3-7.0 for IPSL-CM6A-LR (left) and MPI-ESM1-2-HR (right).

end of the 1500s (peaking in 1600 CE) is exceptional and strikingly in phase with the decreased temperatures inferred from Scandinavian tree rings (Figs. 3, S19), suggesting a reduction in Scandinavian blocking, with less rainfall in Svalbard. These findings are consistent with reconstructed sea-ice extent in the Western Nordic Seas and the Nordic Seas, which indicate that the largest extent was observed during the transition from the 17th to the 19th century and remained high throughout that period[36,40]. During the same period (1600s to the mid-1800s), the local glacier Linnébreen, located 9 km south of Linnévatnet, was at its largest size in the past 2000 years[29]. Beneath Linnébreen lie coal-bearing sandstones, and as the glacier expands, it leads to increased subglacial erosion of the Carboniferous coal[30]. This eroded material is then carried by Linnéelva (the main river) to the lake. Consequently, the accumulation of total organic carbon in Linnévatnet serves as a proxy for monitoring phases characterized by increased glacial activity[29,30]. Thus, increased organics are associated with cooler conditions, and vice versa[29,30]. The organic record clearly shows the advance of Linnébreen during this time of extensive sea-ice and cooler SSTs in the northern Atlantic[41]. The N-Trend Scand tree-ring temperature reconstruction[31] also exhibits a strong negative correlation with organics at Linnévatnet (Fig. S21, $r = -0.63$, $p < 0.001$), indicating that warm anticyclonic conditions in Scandinavia (which favor tree growth) are associated with times of glacier recession in Linnédalen. Hence, this suggests that periods of blocking over Scandinavia and Eurasia were more active in the early LIA (~1400–1570 CE) than from the 1600s to the mid-1800s. After the 1600–1850s cooling period, paleoproxies recording Atlantic water

temperatures in the Fram Strait region, located west of Svalbard, indicate sustained and rapidly warming SSTs from the mid-1800s to the present[42], in line with the steady increase in Ca values recorded for Linnévatnet during that period, which finds parallels with the wider Northern Hemisphere temperature reconstruction[43].

Investigation into the daily to multidecadal hydroclimate variability in Svalbard and associated mechanisms provides context to understand future changes. The exact character of the projected changes will be determined by increases in sea surface and air temperatures in the North Atlantic and changes in the frequency and intensity of Scandinavian and Ural blocking episodes. The persistent rise in SSTs in the Nordic Seas over at least the past ~120 years has been striking (Fig. S22), consistent with the progressive 'Atlantification' along the Fram Strait region that started at the beginning of the 20th century[44]. As sea ice declines and the SSTs continue to increase in the future, the active role of the Nordic Seas as a source of significant moisture cannot be disregarded. A recent analysis of extreme precipitation in Svalbard highlights that sea ice reduction in the Greenland Sea intensifies extreme precipitation on the western coast of Svalbard[35]. Hence, the diminishing or even complete absence of sea ice cover in the Nordic Seas during summer and fall seasons underscores an undeniable reality: the warming trend initiated in the mid-1800s[42] is likely to persist, leading to a subsequent increase in the transportation of moisture into the Svalbard region. Although most climate models project a general reduction in blocking in the northern hemisphere, the Ural Blocking during summer is expected to intensify considerably

with sustained warming[39]. The augmented blocking in the Ural and Scandinavian regions in the future combined with the projected declines in sea ice, and increases in sea surface temperatures and moisture availability, will likely increase the magnitude and frequency of exceptional rainfall incidents similar to the 2016 occurrence, posing further hazards to the population and ecosystems in Svalbard.

## Methods

### Instrumental, reanalysis data and regression maps

The observational data for precipitation between 1955 and 2019 shown in Fig. 1a is based on two gridded products: University of Delaware (UDEL[45]); Global Precipitation Climatology Centre (GPCC[46]). The atmospheric fields—geopotential height at 500 hPa ($Z500$), zonal and meridional winds at 500 hPa ($U500$, $V500$), and surface air temperature (SAT)—for the period 1955–2019 were obtained from the ECMWF Reanalysis v5 (ERA5)[47] product. Sea surface temperature (SST) data were obtained from HadISST[48] and the Extended reconstructed SST (ERSST)[49]. Regression coefficients shown in Figs. 1 and 4 were calculated between the annual mean anomaly fields and standardized time series or indices, indicating a change in a given field for one standard variation change in the time series. The regression analysis and significance tests were conducted in Python using scipy.stats.linregress function from the SciPy library.

### Climate model data

Paleoclimate model simulation data are based on the MPI-ESM1-2-LR *past2k* transient simulation covering the period 1–1850 CE[37], generated as part of the Paleoclimate Model Intercomparison Project Phase 4 (PMIP4). The *past2k* experiment is based on natural forcings over the last two millennia that include evolution of trace gases, volcanic eruptions, solar variability, and land-use. The following fields—SAT, precipitation, Z500, specific humidity and meridional winds at 850hPa—were obtained from the Earth System Grid.

Projections for seasonal and extreme precipitation in Svalbard show in Fig. 5 are based on two CMIP6 climate models: IPSL-CM6A-LR, MPI-ESM1-2-HR. The data for these models were obtained from the Earth System Grid and the Inter-Sectoral Impact Model Intercomparison Project Phase 3b (ISIMIP3b[50,51]) archive, which provides bias-adjusted and statistically downscaled data for five CMIP6 models for studies of climate impacts focusing on the Arctic. The temperature and precipitation projections shown in Fig. 5a are based on raw CMIP6 data for three emissions trajectories described by the Shared Socio-economic Pathways (SSPs). For the analysis of changes in the frequency of extreme precipitation in Svalbard shown in Fig. 5b, we use only one scenario, SSP3-7.0. The SSP3-7.0 scenario is a new forcing pathway included in CMIP6 constructed by combining RCP7.0 with the SSP3 storyline, which together represent the medium to high end of future forcing pathways[52]. The SSP3-7.0, characterized by high emissions, aligns with a plausible projection of how emissions may evolve in the coming decades[53], and is therefore considered a suitable high-end scenario to be used in impact studies[54]. We use only one scenario here since our primary objective is to simply illustrate the direction of change in Svalbard precipitation to increasing emissions in the future.

### Calculation of the blocking index over the Common Era

We calculate the temporal evolution of the SB index over the Common Era by extending the method used over the instrumental period by Lee et al.[38]. The method involves performing the principal component analysis on Z500 anomalies over the domain 60°W to 50°E, 60°N to 85°N. The leading mode of Z500 variability for the period 1–1850 CE describes NAO variations while the second mode shows the SB pattern (described as Scandinavian-Greenland pattern in Lee et al.[38]). The corresponding principal component (PC2) describes the temporal variation of the blocking index over the Common Era and is used in comparisons with other fields shown in Fig. 4.

### Sediment traps at mooring C

Each trap is fixed using brackets and zip ties, spaced 3 m apart, with the basal trap 1 m above the lake floor. A 12 cm diameter funnel, equipped with a baffle cover, delivers sediment to the receiving tube. Timing and thickness of sedimentation events in the grain size profile of the sediment trap from mooring C is correlated to the timing and thickness of sedimentation events recorded in an intervalometer, deployed on a mooring adjacent to mooring C, where the composite sediment core was retrieved. A vertical receiving tube in the intervalometer is equipped with an array of LED lights and corresponding photo diodes that are activated every 30 min. As sediment accumulates vertically in the tubes the LED lights and diodes are blocked. Thus, the timing and amount of vertical accumulation is recorded by the rate by the change in voltage on an Onset HOBO data logger. The intervalometer has been in operation since 2012 except for the year 2014.

### Grain size and μ-XRF from sediment traps at mooring C

The receiving tubes from sediment traps at mooring C were split in two. Grain-size samples were collected in continuous 0.5 cm slices and analyzed using a Beckman Coulter LS 13 320 laser diffraction particle size analyzer after disaggregation in a sodium hexametaphosphate solution. To obtain elemental variability, the other half of the sediment trap was processed using an Itrax X-ray fluorescence core scanner[29].

### Sediment cores at Mooring C

In April 2019, two sediment cores were obtained from mooring C[29]. These cores comprise a UWITEC gravity core measuring 41.8 cm and a piston corer[55] was used to retrieve a sediment core spanning 498 cm in length. The uppermost sediment of the gravity core was carefully stabilized using floral foam to minimize surface disturbance. The longer core was extracted in a single drive and subsequently divided into four segments labeled as follows: LVT19-P2-A (120.8 cm), LVT19-P2-B (120.4 cm), LVT19-P2-C (124 cm), and LVT-P2-D (133 cm). Ninety-four thin sections, each overlapping the other, were made to compile the 5.03-meter composite sequence. Utilizing a custom-made software package 'Analyze Image'[56], the digitalized thin sections were stored and specific regions of interest were selected for acquisition via the scanning electron microscope (SEM). Approximately 5,000 backscattered electron (BSE) images were meticulously extracted to cover the upper 370 cm of the composite core. These SEM images, 8-bit grayscale in nature and sized at 1,024 × 768 pixels, were acquired using an accelerating voltage of 20 kV, maintaining an 8.5 mm working distance and a pixel size of 1 μm. Through a detailed examination of these BSE images, varve boundaries were manually identified and defined[57,58].

### Annually resolved grain size and μ-XRF

The sediment cores retrieved at mooring C were analyzed for their physical and geochemical properties. Annual grain size was extracted from about 5,000 images of thin sections acquired at the Scanning Electron Microscope (SEM) in backscattered mode[29]. Calcium data were obtained using an Itrax X-ray fluorescence core scanner[29], and the annual average was calculated using the same procedure as in Lapointe et al.[41]. Annual grain size and Calcium data cover the upper 370 cm of the composite record where regular laminations are present[29].

### Loss of ignition on the sediments

Loss on ignition (LOI) data[29] involved individual measurement of the 384 crucibles before and after containing the wet sediment. Wet weight was determined by subtracting the mass of the empty crucible from the combined mass of the wet sample and crucible. Subsequently, the same procedure was followed after subjecting the sediment to drying at 60 °C for a period of two days, allowing for the measurement of dry density (achieved by subtracting the mass of the crucible from the dry sample mass). To assess the percentage of organic matter, LOI methodology was employed using a combustion

temperature of 550 °C for a minimum of four hours[59]. The dried samples were then stored in a desiccator, employing Drierite to prevent moisture accumulation. The percentage of organics was determined by subtracting the dry sample mass from the LOI mass. This difference was divided by the dry sample mass and then multiplied by 100 to obtain the percentage of organic matter within the samples.

## Data availability

**Tree-ring data**. The two reconstructed Scandinavian summer temperatures can be found at: https://www.sciencedirect.com/science/article/abs/pii/S0277379117301592. https://www.ncei.noaa.gov/pub/data/paleo/pages2k/pages2k-temperature-v2-2017/data-current-version/Eur-NorthernScandinavia.Esper.2012.txt. **Sea-ice data**. Reconstructed sea ice data from Brennan and Hakim[36] can be accessed at the Zenodo repository: https://zenodo.org/records/5809703. **Linnévatnet data**. The calcium data generated in this study have been deposited in the Arctic data center of the US National Science Foundation: https://arcticdata.io/catalog/view/doi%3A10.18739%2FA2QR4NS2X. All other data are available from the corresponding authors upon request.

## Code availability

The codes generated in this study have been deposited in the Zenodo repository: https://zenodo.org/records/10983058.

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

## Acknowledgements

We are thankful for the financial support provided by grants from the US National Science Foundation (NSF 1744515 and 1744433). Research was supported by grants to the University of Massachusetts (NSF 1744515) and Bates College (NSF 1744433).

## Author contributions

F.L. performed the laboratory work and collected the data from the Linnévatnet's composite sedimentary record. A.V.K. led the analysis and interpretation of data from climate model simulations. F.L. performed spatial correlation maps. M.J.R. conducted field surveys and collected the sediment traps. F.L. led the compilation of the data, interpretation, and writing of the manuscript with R.S.B., A.V.K., M.J.R., and F.W. providing comments and feedback throughout the process.

## Competing interests

The authors declare no competing interests.
