## [Peer Review File · Nature Communications]

Climate extremes in Svalbard over the last two millennia are linked to atmospheric blockingREVIEWER COMMENTS

Reviewer #1 (Remarks to the Author):

This manuscript combines paleodata and instrumental measurements with climate modelling to provide a consistent picture of past, present and future climatic conditions of Svalbard as an example for the Arctic realm. The Arctic is much more exposed to ongoing climatic change with intensified responses of the environmental system threatening (and damaging) the socio-economic infrastructure. As such, the followed approach is unique, forward-looking and very timely to significantly improve our knowledge about Arctic environmental responses to climate change as well as to foster necessary collaborations between paleosciences, climatology and modelling. This approach will set new standards to anticipate the consequences of global change. As such, the work carried out is supportive for all conclusions drawn. This paper is highly recommended to be published in "Nature Communications".

There is only one topic that needs additional attention to increase its understandability for the readership: "organic content/record" is mentioned in the text (lines 284ff) and "organic concentration" in Fig. S18. However, no explanation is provided how this record was created. Generation of this dataset needs to be explained and added to the methods chapters, which should be accomplished during revisions.

In a separate pdf file I make a number of suggestions to improve the readability of the text.

I very much enjoyed reading this manuscript!
Bernd Zolitschka

Reviewer #2 (Remarks to the Author):

General idea:

This paper offers a well-crafted examination of the relationship between Svalbard's precipitation/temperature and the Scandinavia Blocking (Ural Blocking). The paper stands out for its clear explanations and fluid language. What sets this study apart is its unique perspective, as it extends this connection beyond recent decades, delving into sediment data to explore patterns over thousands of years.

In the realm of winter weather patterns, researchers like Yao et al. (2017) have extensively investigated the link between sea ice decreases/extremely warm climate and Scandinavia Blocking over the Barents Sea region. However, the relationship in this research isn't groundbreaking, as it aligns with common

meteorological understanding that blocking usually brings higher temperature and more precipitation to its west wrings.

What piques my interest is the historical variability of the Scandinavia Blocking (SB) index over the last millennium. Upon examining Figure 3a, it becomes evident that temperatures over Scandinavia have risen in recent decades, suggesting an increasing occurrence of SB blocks. However, prior to this period, there appears to be a significant negative trend, which probably suggests a declining SB block frequency. The intriguing correlation between this negative temperature trend and the decline in calcium levels in Figure 3a points to a potential connection between temperature trends and blocking occurrences.

To further improve the study, I have following concerns:

1. It's important to acknowledge that the relationship between Svalbard's temperature and the blocking index isn't particularly robust. Svalbard's location poses a unique challenge for this type of study. It sits at the periphery of the Scandinavia Blocking, making it a complex location to analyze.

During the summer months, the situation differs from what we observe in winters. Inside the blocking, including its edge, it tends to be warmer due to stronger subsidence and higher levels of short-wave irradiance. In contrast, during the winter, it's warmer at the edge of the blocking due to stronger sensible heat flux and longwave irradiance, which are induced by atmospheric rivers.

To make the study more convincing, it would be beneficial to investigate this relationship separately for autumn and summer. While I haven't delved into the Arctic climate during autumn (September to November), based on my previous experiences, I'm confident it could have an impact on your results. Besides, could you also conduct confidence test for the linear regression in Figure 1b, c, d and Figure 3 b, d and Figure 4 a,b?

2. When you extend the current links to the past 1000 years, it has been potentially assumed that the pattern in Figure 4a is the highly dominant pattern for the regional precipitation and warming in Svalbard. This assumption is partially approved by the linear regression of precipitation against 500 hPa geopotential height in Figure 1b. I suggest you to calculate similarly in MPI-ESM1-2_LR the linear regression of precipitation against geopotential height at 500 hPa, which is more closely related to your conclusion that the increased blocking occurrence leads to the increase/decrease of yearly precipitation, resulting in high/low levels of calcium at Linnévatnet. What you have done is relating the temperature and geopotential height first and afterwards calculate the linear correlation between temperature and precipitation, which indirectly links the geopotential height and precipitation. Why not calculate directly the linear regression of Calcium/precipitation against geopotential height.

3. When you analyze the CMIP6 data, you suggest that the increasing blocking occurrence would lead to increasing precipitation in the future, because the global warming. Could you elaborate the potential relation between Scandinavia Blocking and precipitation in the CMIP6 data? For example, linear regression of precipitation against geopotential height at 500 hPa in the climate model IPSL-CM6A-LR or MPI-ESM1-2-HR. Actually you can also calculate the linear trend of Geopotential height at 500 hPa, then you can tell if both SB blocking and precipitation at Svalbard have a positive trend.

4. I do not quite understand why you compare the D50 with Eurasian reconstructed temperature in Figure 3c. For me, Eurasia is a quite huge area including Europe and Asia. What can the correlation of D50 at Linnévatnet and Eurasian reconstructed temperature tell us?

Reference:

Yao, Y., Luo, D., Dai, A., & Simmonds, I. (2017). Increased quasi stationarity and persistence of winter ural blocking and Eurasian extreme cold events in response to arctic warming. Part I: Insights from observational analyses. *Journal of Climate*, 30(10), 3549–3568. <https://doi.org/10.1175/JCLI-D-16-0261.1>

Reviewer #3 (Remarks to the Author):

“Atmospheric blocking drives climate extremes in Svalbard” by Lapointe, F., Karmalkar, A., Bradley, R.S., Retelle M., Wang, F.

In this manuscript, the authors present a set of proxy-based, observational and climate model data to reconstruct climate extremes in Svalbard for the last 1500 yrs and their link to atmospheric blocking patterns. The manuscript shows interesting results which were mostly already published in a recent paper (Lapointe et al. 2023; *Arctic, Antarctic, and Alpine Research*, 55:1, 2223403, DOI: 10.1080/15230430.2023.2223403). The manuscript is well-written and most of the interpretations are consistent. Authors associate local climate extremes with Scandinavian Blocking through the deposition of coarse sediments (rich in Ca) during extreme rainfall events. They analyse the varved sediments from Linnévatne (Svalbard) and confirm this link has been active for the CE. Moreover, they use climate model simulations to support their interpretations and make some future predictions. Based on the presented results, authors suggest an increase of intense rainfall episodes for the current century. I truly believe that the work hypothesis is appropriate, and the conclusions of the manuscript would be promising in terms of impact, however they must be better argued before the MS is being worthy for publication. In other words, I think the MS has potential to be published, but some further clarifications are needed.

Main concerns are about:

i) I miss a reference to chronological framework of the core. Authors do not mention anything about it. How was the age-depth model constructed? Only a reference to a published paper is not enough. Authors should give more details, at least as supplementary information. The reader does not have to go to other papers to get this information. Of course, even if it is already published it deserves a detailed description. Are three radiocarbon dates enough? What about ages between 0 and 750 CE? Are varve counts enough or they need to be supported by other dating methods? These questions need to be clarified.

ii) I wonder if authors are mixing time resolutions. Are they reconstructing extreme daily precipitation patterns or years (Jun-Nov) of extreme precipitation? What about millennial timescale conclusions they state? Authors do not present a millennial timescale record. Why are they saying this? They should make

it very clear through the MS. In my opinion, it is unclear in its current form. They refer to daily data in some parts and then they use extended summer results in other parts. As a paleoclimatologist, I understand this is not easy to solve, and sometimes we tend to go too far with our data. I am flexible in this sense, and I understand it is a difficult issue. For this reason, I ask authors for giving more details and making it clearer.

iii) Authors say that “The principal component analysis of the regional Z500 anomaly field (see Methods) shows that NAO is the leading mode of variability followed by the mode representing the SB pattern (described as Scandinavian-Greenland pattern in Lee et al., (2020)).” However, they do not discuss the role of the NAO. I can agree with them that the NAO do not play a key role in the study site, but authors must discuss it and, more important, explain why and demonstrate it.

iv) Methods are poorly developed, they need to be better explained. This is a publication for a non-specialized journal and methods must be very clear. For example, I miss a detailed explanation of the use of the sediment traps and their correlation with intervalometers. How do authors obtain the data from sediment traps and do they know the age of the samples? or authors mention a composite core, but they do not explain how it was obtained. These are just a couple of examples...

v) CMIP6 model projection of Svalbard precipitation is not correlated and tested with instrumental data for the historical period (1980-2014). Why? Both IPSL and MPI models do not show a good correlation between them, and they do not reflect the extreme rainfall conditions recorded in the sediment record and the instrumental data. Why? If authors do not justify and explain this, they cannot use these model simulations to predict number of days of high precipitation (Fig.5b). Perhaps, the presented model simulations are useful to demonstrate an increasing precipitation trend at decadal-to-centennial timescales, but they are not for daily projections.

vi) Authors correlate extraordinary coarse deposition with strong SB conditions based on reanalysis data. As the authors use reanalysis data instead of station-based data, I wonder why they do not correlate the extraordinary coarse deposition with daily precipitation. For example, authors highlight the 2016 peak of figure 1a, but they do not mention year 2015 or 2017 which also record coarse sediments, but the precipitation is low. The authors should explain this better. In this sense, is there any possibility that coarse sediments are being accumulated due to glacier melting when high temperatures are recorded? I would like to see a bit of discussion on this.

Other minor comments:

Lines 33-34: “Over the past century, temperatures in Svalbard have undergone a concerning increase of approximately 4°C.” Add reference, please.

Line 35: “the already elevated Arctic temperature average” indicate period of the Arctic temperature average.

Lines 42-43: “Svalbard serves as a unique natural laboratory to study drivers of long-term hydroclimate variability and sheds light on the impending changes in the wider Arctic region.”. This is a personal

appreciation but there is an abuse of the use of “unique natural laboratory” in the bibliography, I suggest the authors to avoid its use.

Lines 167-169: Why authors say the Ca is a proxy for heavy rains and then they compare it with temp reconstructions. I can understand the temp and precipitation are correlated and, probably, there are no comparable precipitation records, but authors should clarify this point.

Lines 172-174: I am not sure if $r=0.30$ and $r=0.42$ show a “strong co-variability”.

Lines 283-285: “This is recorded in the organic content of Linnévatnet sediments as the glacier overlies coal-bearing sandstones, so increased organics are associated with cooler conditions, and vice versa.” If the glacier overlies the coal-bearing sandstones how they can reach the lake? Is it related to the ice melting? Explain, please. Moreover, if I look at figure S18, I can see the global correlation is good, but curves only fit well after 1600 CE, before the periods most of the peaks are anti-phased.

Lines 317-319: “Although most climate models project a general reduction in blocking in the northern hemisphere, the Ural Blocking during summer is expected to intensify considerably with sustained warming. The augmented blocking in the Ural and Scandinavian regions in the future...”. Why do authors mix Ural and Scandinavian blockings? This statement only refers to Ural blocking but does not refer to Scandinavian, which is the central focus of the MS. According to Davini and Andrea, (2020), the sector influencing Svalbard would be what they call Eastern-Central Europe. Please, give more details or change these two sentences, the latter is not the consequence of the former.

Lines 330-301: “Investigation into the millennial-scale hydroclimate variability in Svalbard”, authors mention millennial scale hydroclimate variability, what about the daily and annual extremes?

Lines 329-331: “The observational data for precipitation between 1955 and 2019 shown in Fig. 1a is based on two gridded products: University of Delaware (UDEL44); Global Precipitation Climatology Centre (GPCC; Schneider et al., 2021).” I wonder why authors do not use the instrumental records they use in Lapointe et al. (2023). This needs a clarification.

Line 258 and 358: Why do authors use the SSP3-7.0 scenario? This needs a detailed explanation.

Fig. 2c. Authors should indicate what the meaning of the color bands (green, yellow, purple, blue).

Fig 3. Authors should improve the quality of this figure. Resolution is low and it is difficult to see the comparison they try to show in panels a and c.

Supplementary Information: In my opinion, authors should give more information on methods and previous works here. In its current form, it is just a list of figures. Please, add some brief explanations and organize it. It is an important part of the MS that now looks like the authors neglect it.

REVIEWER COMMENTS

Reviewer #1 (Remarks to the Author):

This manuscript combines paleodata and instrumental measurements with climate modelling to provide a consistent picture of past, present and future climatic conditions of Svalbard as an example for the Arctic realm. The Arctic is much more exposed to ongoing climatic change with intensified responses of the environmental system threatening (and damaging) the socio-economic infrastructure. As such, the followed approach is unique, forward-looking and very timely to significantly improve our knowledge about Arctic environmental responses to climate change as well as to foster necessary collaborations between paleosciences, climatology and modelling. This approach will set new standards to anticipate the consequences of global change. As such, the work carried out is supportive for all conclusions drawn. This paper is highly recommended to be published in "Nature Communications". There is only one topic that needs additional attention to increase its understandability for the readership: "organic content/record" is mentioned in the text (lines 284ff) and "organic concentration" in Fig. S18. However, no explanation is provided how this record was created. Generation of this dataset needs to be explained and added to the methods chapters, which should be accomplished during revisions.

In a separate pdf file I make a number of suggestions to improve the readability of the text.

I very much enjoyed reading this manuscript!
Bernd Zolitschka

We thank Dr. Zolitschka for their overall positive assessment of our paper. We have now added information about the organic content of the record in the methods section.

Additionally, we have made revisions to the paper based on Dr. Zolitschka's suggestions. They are highlighted in red in the text.

Reviewer #2 (Remarks to the Author):

General idea:

This paper offers a well-crafted examination of the relationship between Svalbard's precipitation/temperature and the Scandinavia Blocking (Ural Blocking). The paper stands out for its clear explanations and fluid language. What sets this study apart is its unique perspective, as it extends this connection beyond recent decades, delving into sediment data to explore patterns over thousands of years.

In the realm of winter weather patterns, researchers like Yao et al. (2017) have extensively investigated the link between sea ice decreases/extremely warm climate

and Scandinavia Blocking over the Barents Sea region. However, the relationship in this research isn't groundbreaking, as it aligns with common meteorological understanding that blocking usually brings higher temperature and more precipitation to its west wings.

What piques my interest is the historical variability of the Scandinavia Blocking (SB) index over the last millennium. Upon examining Figure 3a, it becomes evident that temperatures over Scandinavia have risen in recent decades, suggesting an increasing occurrence of SB blocks. However, prior to this period, there appears to be a significant negative trend, which probably suggests a declining SB block frequency. The intriguing correlation between this negative temperature trend and the decline in calcium levels in Figure 3a points to a potential connection between temperature trends and blocking occurrences.

To further improve the study, I have following concerns:

1. It's important to acknowledge that the relationship between Svalbard's temperature and the blocking index isn't particularly robust. Svalbard's location poses a unique challenge for this type of study. It sits at the periphery of the Scandinavia Blocking, making it a complex location to analyze.

During the summer months, the situation differs from what we observe in winters. Inside the blocking, including its edge, it tends to be warmer due to stronger subsidence and higher levels of short-wave irradiance. In contrast, during the winter, it's warmer at the edge of the blocking due to stronger sensible heat flux and longwave irradiance, which are induced by atmospheric rivers.

To make the study more convincing, it would be beneficial to investigate this relationship separately for autumn and summer. While I haven't delved into the Arctic climate during autumn (September to November), based on my previous experiences, I'm confident it could have an impact on your results. Besides, could you also conduct confidence test for the linear regression in Figure 1b, c, d and Figure 3 b, d and Figure 4 a,b?

There are two questions here: first is about the robustness of the SB-Svalbard temperature relationship across seasons and the second is about the statistical significance of the regression maps presented in the paper. To address the second question, we have added cross-hatching to our Figs. 1 and 4, indicating regions with statistically significant relationships at the 95% level. These results clearly show that the relationships are statistically robust for regions under consideration.

With regard to the first question about seasonal dependence:

We agree with the reviewer that the link between Svalbard's temperature and Scandinavian blocking is season-dependent, although temperature does generally increase when the SB index gets higher. If we consider all rainfall events greater than 10mm during winter (Nov. to March) since 1958, it is clear that an atmospheric pattern similar to Scandinavian and/or Urals Blocking was present during those events. We have added Table S1 listing the days of these large rain events and Figures S2 & S3 showing the atmospheric anomalies during those days.

Additionally, the following figure shows regressions between Svalbard precipitation and z500 and SAT for summer (JJA) and fall (SON) separately. While the centers of action are in slightly different positions, blocking-type patterns are associated with Svalbard precipitation anomalies in both seasons.

Figure caption: The relationships between Svalbard precipitation and Z500 and SAT for JJA and SON separately. Same as Figs. 1b and 1d in the manuscript.

Thus, the instrumental evidence shows that the same mechanism is at play for triggering high rain events in Svalbard both in winter and summer. However, as stated in the main text our study is focused on the June to November months as there is no sediment input that can be recorded in other months because of the ice covering the lake (lines 131-133). We added a sentence (lines 111-113) to let know the reader that atmospheric mechanism is similar regardless of the seasons.

2. When you extend the current links to the past 1000 years, it has been potentially assumed that the pattern in Figure 4a is the highly dominant pattern for the regional precipitation and warming in Svalbard. This assumption is partially approved by the linear regression of precipitation against 500 hPa geopotential height in Figure 1b. I suggest you to calculate similarly in MPI-ESM1-2_LR the linear regression of precipitation against geopotential height at 500 hPa, which is more closely related to

your conclusion that the increased blocking occurrence leads to the increase/decrease of yearly precipitation, resulting in high/low levels of calcium at Linnévatnet. What you have done is relating the temperature and geopotential height first and afterwards calculate the linear correlation between temperature and precipitation, which indirectly links the geopotential height and precipitation. Why not calculate directly the linear regression of Calcium/precipitation against geopotential height.

We have modified Fig. 4 to include the regression pattern between the Scandinavian Blocking (SB) index (second mode of z500 variability) and regional precipitation as the reviewer suggested. It does show higher regional precipitation in Svalbard area corresponding to high SB index.

The analysis presented in Fig. 4 is slightly different to the observational-based analysis shown in Fig. 1b, in which we regressed Svalbard precipitation with regional z500 anomalies. In the case of the model analysis, we first identified the SB regime using the EOF analysis and studied its impact by regressing the SB index with Arctic-wide variables (Fig. 4) to examine processes the regionwide impact of the blocking pattern.

Additionally, it's important to note that a direct proxy-model comparison is tricky. Since models have their own flavor of internal climate variability, we do not expect modeled variability to exactly match that of proxy data, especially the high frequency variations that are not necessarily externally forced. Therefore, we do not compare Calcium (proxy) and model data directly but instead test our proposed mechanism using model data alone. That is, we show that the MPI past2k paleoclimate simulation clearly shows a connection between Scandinavian blocking and regional temperatures and precipitation using model data alone.

3. When you analyze the CMIP6 data, you suggest that the increasing blocking occurrence would lead to increasing precipitation in the future, because the global warming. Could you elaborate the potential relation between Scandinavia Blocking and precipitation in the CMIP6 data? For example, linear regression of precipitation against geopotential height at 500 hPa in the climate model IPSL-CM6A-LR or MPI-ESM1-2-HR. Actually you can also calculated the linear trend of Geopotential height at 500 hPa, then you can tell if both SB blocking and precipitation at Svalbard have a positive trend.

A full exploration of the connection between Scandinavian Blocking and precipitation in CMIP6 models is not included in this study for the following reasons.

First, while we can certainly show trends in z500 and precipitation as the reviewer suggests, establishing a clear dynamically link between the two in CMIP6 models will still require considerable work involving identifying blocking regimes in the models, assessing their credibility against observations, and establishing a causal link with Svalbard precipitation. This will require careful assessment of all thermodynamic (moisture increase with warming) and dynamical (circulation changes) factors that will drive precipitation variability in Svalbard in the future.

Second, the representation of blocking in the current generation of models and projected changes in blocking under future conditions is still an active area of research with considerable uncertainties. Instead, we limit our analysis to a couple of CMIP6 models to simply illustrate that model projections indicate a steady increase in seasonal precipitation as well as an increase in the frequency of precipitation (number of days with precipitation above 10 mm). We further cite previous studies (Hanna et al., 2018; Davini and d’Andrea, 2020) that discuss modeled changes in blocking episodes under modern and enhanced GHG conditions in the future. These studies discuss regional consequences of blocking and do not specifically focus on precipitation in Svalbard. A full analysis of blocking episodes in CMIP6 models will likely require another study since models vary substantially in their ability to capture blocking trends and frequencies (see Davinin and d’Andrea, 2020 cited in the manuscript) and regional precipitation.

We have added a couple of sentences to clarify our approach (Lines 290-299).

4. I do not quite understand why you compare the D50 with Eurasian reconstructed temperature in Figure 3c. For me, Eurasia is a quite huge area including Europe and Asia. What can the correlation of D50 at Linnévatnet and Eurasian reconstructed temperature tell us?

This is a good point. We decided to remove D50 from the main text and to replace it by a comparison between Ca and reconstructed sea ice from the Nordic Seas as this area is of paramount importance for moisture transport towards Svalbard. We think that it improved the manuscript and adds support to our hypothesis.

Reference:

Yao, Y., Luo, D., Dai, A., & Simmonds, I. (2017). Increased quasi stationarity and persistence of winter Ural blocking and Eurasian extreme cold events in response to arctic warming. Part I: Insights from observational analyses. *Journal of Climate*, 30(10), 3549–3568. <https://doi.org/10.1175/JCLI-D-16-0261.1>

Thanks, we added this reference.

Reviewer #3 (Remarks to the Author):

“Atmospheric blocking drives climate extremes in Svalbard” by Lapointe, F., Karmalkar, A., Bradley, R.S., Retelle M., Wang, F.

In this manuscript, the authors present a set of proxy-based, observational and climate model data to reconstruct climate extremes in Svalbard for the last 1500 yrs and their link to atmospheric blocking patterns. The manuscript shows interesting results which were mostly already published in a recent paper (Lapointe et al. 2023; *Arctic, Antarctic, and Alpine Research*, 55:1, 2223403, DOI: 10.1080/15230430.2023.2223403). The

manuscript is well-written and most of the interpretations are consistent. Authors associate local climate extremes with Scandinavian Blocking through the deposition of coarse sediments (rich in Ca) during extreme rainfall events. They analyse the varved sediments from Linnévatne (Svalbard) and confirm this link has been active for the CE. Moreover, they use climate model simulations to support their interpretations and make some future predictions. Based on the presented results, authors suggest an increase of intense rainfall episodes for the current century. I truly believe that the work hypothesis is appropriate, and the conclusions of the manuscript would be promising in terms of impact, however they must be better argued before the MS is being worthy for publication. In other words, I think the MS has potential to be published, but some further clarifications are needed.

Main concerns are about:

i) I miss a reference to chronological framework of the core. Authors do not mention anything about it. How was the age-depth model constructed? Only a reference to a published paper is not enough. Authors should give more details, at least as supplementary information. The reader does not have to go to other papers to get this information. Of course, even if it is already published it deserves a detailed description. Are three radiocarbon dates enough? What about ages between 0 and 750 CE? Are varve counts enough or they need to be supported by other dating methods? These questions need to be clarified.

We agree with the reviewer about the lack of information on the chronology of this varve record. We added text in the supplementary materials (Supplementary text 1) on the varve counts and the chronological constraints. It is really challenging to find any suitable materials to date in this clastic and high-arctic environment, while dating of bulk sediment generally gives unreliable ages (too old because of the old carbon stored in watersheds). This is why we use the macro-fossils and their ^{14}C ages from a sediment core that receive twice the sedimentation rates compared to our composite sequence. Thanks to an excellent match in the chemo-stratigraphy from that core and our core we were able to locate the exact locations of these macrofossils and compare their ^{14}C dates to our varve counts which reveal a good match. It is true that there is no age control between 1 and 750 CE, we explain in the supplementary text 1 that sedimentation rates do not change much during that time interval, and that the section is likely to be annually laminated as well.

ii) I wonder if authors are mixing time resolutions. Are they reconstructing extreme daily precipitation patterns or years (Jun-Nov) of extreme precipitation? What about millennial timescale conclusions they state? Authors do not present a millennial timescale record. Why are they saying this? They should make it very clear through the MS. In my opinion, it is unclear in its current form. They refer to daily data in some parts and then they use extended summer results in other parts. As a paleoclimatologist, I understand this is not easy to solve, and sometimes we tend to go too far with our data. I am flexible in this sense, and I understand it is a difficult issue. For this reason, I ask authors for giving more details and making it clearer.

If we simply average over a seasonal basis (summer/winter/fall) we always get the same pattern of blocking. Analysis using daily precipitation also show the same pattern (Figs 2, S2, S3) hence whether it is monthly or daily anomalies, the same atmospheric dynamics are at play : Scandinavian blocking and/or Urals Blocking. We agree with the reviewer that we should explain and make that point better. We added a sentence in line 145-147.

iii) Authors say that “The principal component analysis of the regional Z500 anomaly field (see Methods) shows that NAO is the leading mode of variability followed by the mode representing the SB pattern (described as Scandinavian-Greenland pattern in Lee et al., (2020)).” However, they do not discuss the role of the NAO. I can agree with them that the NAO do not play a key role in the study site, but authors must discuss it and, more important, explain why and demonstrate it.

There are two primary reasons why we do not discuss NAO in detail in this study:

Reason 1: The regression analysis based on data over the historical period clearly indicates the role of blocking-type circulation on Svalbard precipitation (see Fig. 1b) and not of NAO. Many other studies do suggest the influence of NAO, especially in winter, through changes in North Atlantic SSTs, sea ice, and low-level circulation (e.g., Hanssen-Bauer et al., 2019), but our focus here is on the warm season precipitation (and wet extremes), which is mainly driven by the SB-type circulation. Additionally, while the SB pattern emerges as the second mode of regional z500 variability it is almost as important as the first mode representing NAO circulation. These first two modes – NAO and SB – explain a large fraction (65.8%) of the total z500 variability; the first mode explains 36.6% while the second mode representing SB explains 29.2% of the total z500 variability.

Reason 2: To further illustrate this, we calculated the regressions between the NAO index (first mode of z500 variability) and regional z500, temperature, and precipitation anomalies (see Figure below). More importantly, panel c does not suggest any relationship between NAO and Svalbard precipitation, mainly because the flow associated with NAO is zonal in nature and lies to the south of Svalbard.

Figure caption: Relationship between NAO and regional temperature and precipitation in MPI-ESM1-2-LR transient simulation over the Common Era (past2k experiment). The analysis is based on the PCA of regional Z500 as described in the manuscript. (a) NAO spatial pattern based on the first mode of Z500 variability. (b) Regressions between regional temperatures and the NAO index. (c) Regressions between regional precipitation and the NAO index.

Hanssen-Bauer, I., Førland, E. J., Hisdal, H., Mayer, S., Sandø, A. B., & Sorteberg, A. (2019). Climate in Svalbard 2100. *A knowledge base for climate adaptation*.

iv) Methods are poorly developed, they need to be better explained. This is a publication for a non-specialized journal and methods must be very clear. For example, I miss a detailed explanation of the use of the sediment traps and their correlation with intervalometers. How do authors obtain the data from sediment traps and do they know the age of the samples? or authors mention a composite core, but they do not explain how it was obtained. These are just a couple of examples...

We have now added and clarified information about the sediment traps, the sediment composite, thin-sections, as well as loss on ignition.

v) CMIP6 model projection of Svalbard precipitation is not correlated and tested with instrumental data for the historical period (1980-2014). Why? Both IPSL and MPI models do not show a good correlation between them, and they do not reflect the extreme rainfall conditions recorded in the sediment record and the instrumental data. Why? If authors do not justify and explain this, they cannot use these model simulations to predict number of days of high precipitation (Fig.5b). Perhaps, the presented model simulations are useful to demonstrate an increasing precipitation trend at decadal-to-centennial timescales, but they are not for daily projections.

With regards to 'testing with instrumental data', note that the output from two CMIP6 models used to provide projections is already bias adjusted to be consistent with observations. The two models come from a set of five carefully selected CMIP6 models recommended for use in studies focusing on the Arctic as part of the ISIMIP exercise (see figure below). Therefore, there is no further need to do model validation. We

specifically pick these two models from the five provided (shown by + signs) because they represent a wide range in temperature and precipitation projections for the Arctic as seen in the figure below. We have included this rationale in the ‘Climate model data’ section under *Methods*.

Figure caption: Temperature and precipitation projections for the Arctic based on CMIP6 models. While the full CMIP6 ensembles spans a wide range in projections, the two models used in this study – MPI-ESM2-HR and IPSL-CM6A-LR – are selected to highlight similarity in their projections for Svalbard despite having different magnitude of temperature and precipitation changes for the Arctic as a whole. Source: <https://gcmeval.met.no/>

The lack of correlation between the models is due to models exhibiting their own flavor of internal variability. In fact, in such model analyses, we do not expect different climate models to show correlations at interannual timescale, especially for local precipitation. While the models do not match exactly in terms of their interannual variations, they both show a long-term increase in projected precipitation in Svalbard, which is what we intend to highlight through Fig. 5.

We agree with the reviewer that the analysis presented in Fig. 5a, as is, cannot be used to argue a projected increase in ‘daily precipitation’. That is precisely the reason why we analyze the change in the number of days with precipitation above 10mm (Fig. 5b). This metric indicates a change in the frequency of extreme precipitation.

In sum, Fig. 5a indicates a projected, long-term increase in Svalbard in months from June through November and Fig. 5b suggest that an increase in the frequency of daily precipitation in the region.

vi) Authors correlate extraordinary coarse deposition with strong SB conditions based on reanalysis data. As the authors use reanalysis data instead of station-based data, I wonder why they do not correlate the extraordinary coarse deposition with daily precipitation. For example, authors highlight the 2016 peak of figure 1a, but they do not mention year 2015 or 2017 which also record coarse sediments, but the precipitation is low. The authors should explain this better. In this sense, is there any possibility that coarse sediments are being accumulated due to glacier melting when high temperatures are recorded? I would like to see a bit of discussion on this.

The primary source of water for Linnéelva (the main river system draining into the lake) originates predominantly from the melting snow and glaciers of Linnébreen, a small valley glacier situated 7 kilometers to the south of Linnévatnet. Hence, the melting of the local glacier Linnebreen, as a consequence of warming, contribute to the accumulation of coarse sediments. But this glacier has reduced in size considerably since the end of the Little Ice Age, and is probably now the thinnest it has been over the last few millennia (Lapointe et al. 2023). So, if we only take into account sedimentation coming from the glacier, it appears very likely that the grain-size will decrease in the future as a result of the shrinking (and the disappearance) of the glacier. In addition to being a temperature (driving the glacier fluctuations) proxy, the grain-size data is also influenced by large rain events whereby increased rainfall induce coarser sediments (Lapointe et al. 2023). In this context and because our focus is on large rain events, we opt for the Calcium data as a better proxy for rainfall events. Indeed, carbonates only exist on the eastern valley, and observations showed that the transport of materials in this area is mainly trigger by rainfall events. In addition, samples collected around the lake's watershed have shown that positive μ -XRF Ca values were only found in this carbonate platform (Lapointe et al. 2023). In contrast to D50 grain-size, calcium is more prone to being recorded in the lake sediments regardless of variations in the glacier's size. We contend that calcium serves as a superior indicator for rainfall compared to D50. We added Figure S17 to show that grain-size response to medium to large rain events (10-20mm) is not as perceptible as the calcium data, and lines 169-184 to provide discussion on this. As per the suggestion from Reviewer 2, we've opted to omit the D50 grain-size data from Figure 3.

Line 35: "the already elevated Arctic temperature average" indicate period of the Arctic temperature average.

"Particularly alarming is the fact that **since 1991**, this region has experienced a warming trend that surpasses the already elevated Arctic temperature average more than twofold"

Since 1991

Lines 42-43: “Svalbard serves as a unique natural laboratory to study drivers of long-term hydroclimate variability and sheds light on the impending changes in the wider Arctic region.”. This is a personal appreciation but there is an abuse of the use of “unique natural laboratory” in the bibliography, I suggest the authors to avoid its use.

Ok, we replaced ‘a unique’ by ‘an ideal’ line 46.

Lines 167-169: Why authors say the Ca is a proxy for heavy rains and then they compare it with temp reconstructions. I can understand the temp and precipitation are correlated and, probably, there are no comparable precipitation records, but authors should clarify this point.

We have added lines 169-184 to better discuss how Ca is a particularly good proxy for large rainfall in Linnévatnet. We have additionally incorporated Figure S19 to illustrate the sensitivity of Ca in response to significant rainfall events, as inferred from both instrumental data and sediment trap observations. The Ca is compared with Scandinavian temperature that tracks high-pressure system over Scandinavia (Fig. 3b). We also added a reconstructed sea ice from the Nordic Seas to give support to the present-day observations (Fig. 1c).

Lines 172-174: I am not sure if $r=0.30$ and $r=0.42$ show a “strong co-variability”.

We have replaced ‘strong’ with significant, line 196.

Lines 283-285: “This is recorded in the organic content of Linnévatnet sediments as the glacier overlies coal-bearing sandstones, so increased organics are associated with cooler conditions, and vice versa.” If the glacier overlies the coal-bearing sandstones how they can reach the lake? Is it related to the ice melting? Explain, please. Moreover, if I look at figure S18, I can see the global correlation is good, but curves only fit well after 1600 CE, before the periods most of the peaks are anti-phased.

We added 3 sentences on lines 322-328 to better make our point of the coal-bearing sandstones and its link to the lake sediments.

It is true that there are few instances where the Scand temp reconstruction and organics show discrepancies, but both time series exhibit notable stepwise declines (increase) for the past ~1250 years. It is particularly evident at the onset of the 1600s. It is possible to identify sudden alterations or shifts using change point analysis. We defined "maximum change point" to establish a limit on the number of potential breakpoints or instances where noteworthy changes in the data pattern are anticipated. We set here a maximum change point to 3, so the analysis will seek up to 3 specific moments within the time series where substantial alterations in behavior might have taken place. The overall coherence in the change point analysis suggests that the data are linked. In addition, when we average both datasets on selected periods of positive (750-1175C; 1380-1580CE) and negative (1176-1380CE; 1580-1850CE) periods they show similar values indicating a

good correspondence on centennial scale. In brief, organics data are low resolution, so we think that this relationship is somewhat worth mentioning.

Figure caption: In red is organics from Linnevatnet, in green the scand temp reconstruction.

As Figure S21, but showing the main periods of positive and negative values being averaged for each datasets.

Lines 317-319: “Although most climate models project a general reduction in blocking in the northern hemisphere, the Ural Blocking during summer is expected to intensify considerably with sustained warming. The augmented blocking in the Ural and Scandinavian regions in the future...”. Why do authors mix Ural and Scandinavian blockings? This statement only refers to Ural blocking but does not refer to Scandinavian, which is the central focus of the MS. According to Davini and Andrea, (2020), the sector influencing Svalbard would be what they call Eastern-Central Europe. Please, give more details or change these two sentences, the latter is not the consequence of the former.

It is true that the main focus is the Scandinavian Blocking, but there are instances where the blocking over the Urals also lead to extreme rain events in Svalbard as seen in the daily geopotential 500hPa heights anomalies (Fig. S2a,b, i, S3c,e, i). When the August to October months is considered, we can see from the spatial correlation between observational precipitation data and Z500 that the center of the blocking extends from Scandinavia through western Siberia, which includes at the center the Urals.

Figure caption: Spatial correlation between airport precipitation at Longyearbyen and the 500hPa anomalies through August-October (1950-2018).

In fact, the center of the blocking for the Ural is shifted east, not too different from that of the Scandinavian blocking (Peings 2019; see Fig. 1a).

Peings Y. 2019 Ural Blocking as a Driver of Early-Winter Stratospheric Warming. GRL

Lines 330-301: “Investigation into the millennial-scale hydroclimate variability in Svalbard”, authors mention millennial scale hydroclimate variability, what about the daily and annual extremes?

We changed millennial-scale for daily to multidecadal (as seen in the instrumental and paleo data). Line 340

Lines 329-331: “The observational data for precipitation between 1955 and 2019 shown in Fig. 1a is based on two gridded products: University of Delaware (UDEL44); Global Precipitation Climatology Centre (GPCC; Schneider et al., 2021).” I wonder why authors

do not use the instrumental records they use in Lapointe et al. (2023). This needs a clarification.

We note that the instrumental record from Longyearbyen (Nordli et al 1996) used in Lapointe et al. (2023) led to similar results as shown below; similar atmospheric blocking pattern. In our previous study, this instrumental record was based on one single location. We believe that the use of gridded product encompassing a wider area of the western Spitsbergen in Svalbard (as in Fig.1) is better suited to capture the teleconnection and associated climatic impacts in this study. We added a sentence in the methods that the use of GPCC and UDEL44 led to similar results as the instrumental datasets from Longyearbyen.

Figure caption: The relationship between Longyearbyen’s airport (LYR; the capital of Svalbard) precipitation and geopotential height at 500hPa (Z500) average over 6 months.

In brief, we use two well-established gridded precipitation products that allow us to determine the robustness of instrumental data and helps us study precipitation for the entire region (as shown in Fig. 1) instead of a point location.

Line 258 and 358: Why do authors use the SSP3-7.0 scenario? This needs a detailed explanation.

We have added the following explanation to the ‘Climate model data’ section under *Methods* (Line 396-408).

“The SSP3-7.0 is a new scenario is a new forcing pathway included in CMIP6 constructed by combining RCP7.0 with the SSP3 storyline, which together represent the medium to high end of future forcing pathways (O’Neill et al., 2016). The SSP3-7.0, characterized by high emissions, aligns with a plausible projection of how emissions may evolve in the coming decades (Riahl et al., 2017), and is therefore considered a suitable high-end scenario to be used in impact studies (Hausfather and Peters, 2020). We use only one scenario here since our primary objective is to simply illustrate the direction of change in Svalbard precipitation to increasing emissions in the future.”

O’Neill, B. C., Tebaldi, C., Van Vuuren, D. P., Eyring, V., Friedlingstein, P., Hurtt, G., ... & Sanderson, B. M. (2016). The scenario model intercomparison project (ScenarioMIP) for CMIP6. *Geoscientific Model Development*, 9(9), 3461-3482.

Riahi, K., Van Vuuren, D. P., Kriegler, E., Edmonds, J., O’neill, B. C., Fujimori, S., ... & Tavoni, M. (2017). The Shared Socioeconomic Pathways and their energy, land use, and greenhouse gas emissions implications: An overview. *Global environmental change*, 42, 153-168.

Hausfather, Z., & Peters, G. P. (2020). Emissions—the ‘business as usual’ story is misleading. *Nature*, 577(7792), 618-620.

Fig. 2c. Authors should indicate what the meaning of the color bands (green, yellow, purple, blue).

Done. Thanks.

Fig 3. Authors should improve the quality of this figure. Resolution is low and it is difficult to see the comparison they try to show in panels a and c.

We have changed that figure and removed the D50 for a comparison of Ca with reconstructed sea ice in the Nordic Seas. We think it gives much more credibility to our findings now.

Supplementary Information: In my opinion, authors should give more information on methods and previous works here. In its current form, it is just a list of figures. Please, add some brief explanations and organize it. It is an important part of the MS that now looks like the authors neglect it.

We have now added two supplementary texts, and a more detailed text in the captions.

REVIEWER COMMENTS

Reviewer #2 (Remarks to the Author):

I am very satisfied with the revised manuscripts excluding a minor suggestion. I have personally learned some new knowledge about the reconstruction of atmospheric circulations. Many thanks!

Minor

I appreciate the inclusion of sea ice concentration in the discussion. However, I have a different perspective on the relationship between sea ice concentration and precipitation at Svalbard. The current text suggests that larger open water to the south of Svalbard favors precipitation. Based on my experience, if we were to analyze the linear relationship between Svalbard precipitation and sea ice concentration to the north of Svalbard, I am confident that we would get a negative linear correlation as well, similar to what Figure 3c illustrates.

My viewpoint is founded on the following considerations. The occurrence of blocking over the Barents Sea, situated to the west of the blocking, tends to bring warm air from lower latitudes into the Arctic. This influx of warmer air can result in increased precipitation at Svalbard while simultaneously decreasing sea ice concentration in that region. While I acknowledge the factors mentioned in your text, such as the potential enhancement of precipitation by larger open water to the south of Svalbard, I believe that the former factor holds more significant and robust sway in sea ice variation.

I kindly suggest adjusting the discussion to incorporate this perspective as a minor modification. Additionally, I would like to recommend a paper authored by me (not for citation) that could provide you with a deeper understanding of the relationship between sea ice, temperature, and blocking.

The Role of Atmospheric Blocking in Regulating Arctic Warming

<https://agupubs.onlinelibrary.wiley.com/doi/10.1029/2022GL097899>

Reviewer #3 (Remarks to the Author):

Authors present an improved version of the previous manuscript, they have clarified most of my previous concerns, which is appreciated. However, there are still two points should be addressed before I can consider it worthy for publication in Nature Communications.

I. Related to the answer to my previous comment: *CMIP6 model projection of Svalbard precipitation is not correlated and tested with instrumental data for the historical period (1980-2014). Why? Both IPSL and MPI models do not show a good correlation between them, and they do not reflect the extreme rainfall conditions recorded in the sediment record and the instrumental data. Why? If authors do not justify and explain this, they cannot use these model simulations to predict number of days of high precipitation (Fig.5b). Perhaps, the presented model simulations are useful to demonstrate an increasing precipitation trend at decadal-to-centennial timescales, but they are not for daily projections.* Authors present a figure with a plot including CMIP6 ensembles for a SSP5 8.5 scenario, however authors do not use this scenario for its projections. In my opinion, authors should include more than one scenario for its future projection. I strongly suggest including at least three of them (e.g., SSPS1-2.6; SSPS2-4.5; SSPS3-7.0 or SSPS5-8.5)

II. I still have my concerns about the use of Scandinavian Blocking and Ural Blocking at almost the same level. Authors should clarify this through the text, as they have done in the rebuttal letter, otherwise they should avoid mentioning the Ural Blocking and the associated reference (*Davini and Andrea, 2020; J. of Climate*), which indicates the reverse pattern for the Scandinavian region.

Reviewer #2 (Remarks to the Author):

I am very satisfied with the revised manuscripts excluding a minor suggestion. I have personally learned some new knowledge about the reconstruction of atmospheric circulations. Many thanks!

We thank the reviewer for all their comments, and glad to hear that they have found the paper insightful. All of the changes made in the main text are in green.

Minor

I appreciate the inclusion of sea ice concentration in the discussion. However, I have a different perspective on the relationship between sea ice concentration and precipitation at Svalbard. The current text suggests that larger open water to the south of Svalbard favors precipitation. Based on my experience, if we were to analyze the linear relationship between Svalbard precipitation and sea ice concentration to the north of Svalbard, I am confident that we would get a negative linear correlation as well, similar to what Figure 3c illustrates.

We do indeed find a negative correlation with reconstructed sea-ice north of Svalbard as well as in the Barents Sea area, however not as strong as the one in the Nordic Seas. In the paper, we focus on sea ice concentrations to the south of Svalbard because that is relevant to our discussion about the southerly flow of atmospheric moisture steered by SB/Ural blocking episodes toward Svalbard.

My viewpoint is founded on the following considerations. The occurrence of blocking over the Barents Sea, situated to the west of the blocking, tends to bring warm air from lower latitudes into the Arctic. This influx of warmer air can result in increased precipitation at Svalbard while simultaneously decreasing sea ice concentration in that region. While I acknowledge the factors mentioned in your text, such as the potential enhancement of precipitation by larger open water to the south of Svalbard, I believe that the former factor holds more significant and robust sway in sea ice variation.

I kindly suggest adjusting the discussion to incorporate this perspective as a minor modification. Additionally, I would like to recommend a paper authored by me (not for citation) that could provide you with a deeper understanding of the relationship between sea ice, temperature, and blocking.

The Role of Atmospheric Blocking in Regulating Arctic Warming

<https://agupubs.onlinelibrary.wiley.com/doi/10.1029/2022GL097899>

We agree with the reviewer that the blocking anomaly (and its upward trend) is the main trigger for sea ice decline in the higher latitudes (including south of Svalbard). We now better clarify that in the main text (lines 317-319). We believe that when the background condition is warmer than normal (like today) the presence of open waters south of Svalbard (and in the Greenland seas) do amplify the potential of higher precipitation in Svalbard (See publication below).

<https://www.sciencedirect.com/science/article/pii/S221209472200024X>

Reviewer #3

Authors present an improved version of the previous manuscript, they have clarified most of my previous concerns, which is appreciated. However, there are still two points should be addressed before I can consider it worthy for publication in Nature Communications.

- I. Related to the answer to my previous comment: *CMIP6 model projection of Svalbard precipitation is not correlated and tested with instrumental data for the historical period (1980- 2014). Why? Both IPSL and MPI models do not show a good correlation between them, and they do not reflect the extreme rainfall conditions recorded in the sediment record and the instrumental data. Why? If authors do not justify and explain this, they cannot use these model simulations to predict number of days of high precipitation (Fig.5b). Perhaps, the presented model simulations are useful to demonstrate an increasing precipitation trend at decadal-to- centennial timescales, but they are not for daily projections.* Authors present a figure with a plot including CIMP6 ensembles for a SSP5 8.5 scenario, however authors do not use this scenario for its projections. In my opinion, authors should include more than one scenario for its future projection. I strongly suggest including at least three of them (e.g., SSPS1-2.6; SSPS2-4.5; SSPS3-7.0 or SSPS5-8.5)

We have modified 'panel a' in Fig. 5 to show both the temperature and precipitation projections for Svalbard this century in two climate models and three SSP scenarios (SSP2-4.5, SSP3-7.0, SSP5-8.5) as requested by the reviewer. The projections are shown as a total change between 2015-2099 calculated from linear trends. This panel replaces precipitation time series shown earlier for just one SSP (SSP3-7.0). We have modified the analysis period in panel b as well to match the period in panel a. These minor modifications do not change any of our conclusions, but only bolster our main agreement that if warming were to continue in the future, model projections indicate a wetter climate for Svalbard this century, which is the sole motivation behind presenting this analysis. All of the changes made in the main text are in green.

Figure 5. Climate projections for Svalbard based on two CMIP6 models. a, Total change in Jun-Nov mean temperature and precipitation in Svalbard (see inset for area definition) between 2015 and 2099 based on linear trends for three emissions scenarios (SSP2-4.5, SSP3-7.0, SSP5-8.5). **b**, Histograms showing the distributions of number of days with precipitation > 10 mm across Svalbard (at 0.5° spatial resolution) over the period 2015-2034 (dashed line) and 2080-2099 (solid line) under SSP3-7.0 for IPSL-CM6A-LR (left) and MPI-ESM1-2-HR (right).

We have modified the text describing Fig. 5 and the figure caption. The modified text makes it clear that this analysis simply illustrates the anticipated changes in the future, but further work is necessary to be able to make robust connections between future changes in extreme rainfall in Svalbard and atmospheric blocking. We also made minor modifications to the methods section to reflect changes made to Fig. 5.

The italicized part of the reviewer comment above was the question they had raised earlier, and our first revision contains a detailed response.

II. I still have my concerns about the use of Scandinavian Blocking and Ural Blocking at almost the same level. Authors should clarify this through the text, as they have done in the rebuttal letter, otherwise they should avoid mentioning the Ural Blocking and the associated reference (*Davini and Andrea, 2020; J. of Climate*), which indicates the reverse pattern for the Scandinavian region.?

We decided to not omit the Ural Blocking because observational evidence shows it can trigger intense rain events in Svalbard. We added lines 321-327 to better explain this as well as Fig. S21 showing that the blocking during SON is more widespread (compared to summer) and includes the Urals near its center.

Figure S21. The relationships between Svalbard precipitation and Z500 and SAT for JJA and SON separately. Same as Figs. 1b and 1d in the main text.